# Ultrasensitive Detection of Cu(II) and Pb(II) Using a Water-Soluble Perylene Probe

**DOI:** 10.3390/molecules27207079

**Published:** 2022-10-20

**Authors:** Erika Kozma, Antonella Caterina Boccia, Anita Andicsova-Eckstein, Alfio Pulvirenti, Chiara Botta

**Affiliations:** 1Consiglio Nazionale delle Ricerche, Istituto per le Scienze e Tecnologie Chimiche ‘Giulio Natta’, Via Alfonso Corti 12, 20133 Milano, Italy; 2Slovak Academy of Sciences, Polymer Institute, Dubravska Cesta 9, 81107 Bratislava, Slovakia

**Keywords:** metal ion sensor, water-soluble perylene, optical absorption, NMR experiments

## Abstract

The selective detection of metal ions in water, using sustainable detection systems, is of crescent importance for monitoring water environments and drinking water safety. One of the key elements of future chemical sciences is the use of sustainable approaches in the design of new materials. In this study, we design and synthesize a low-cost, water-soluble potassium salt of 3,4,9,10-perylene tetracarboxylic acid (PTAS), which shows a selective optical response on the addition of Cu^2+^ and Pb^2+^ ions in aqueous solutions. By using a water-soluble chromophore, the interactions with the metal ions are definitely more intimate and efficient, with respect to standard methods employing cosolvents. The detection limits of PTAS for both Cu^2+^ and Pb^2+^ are found to be 2 µM by using a simple absorbance mode, and even lower (1 μM) with NMR experiments, indicating that this analyte–probe system is sensitive enough for the detection of copper ions in drinking water and lead ions in waste water. The complexation of PTAS with both ions is supported with NMR studies, which reveal the formation of new species between PTAS and analytes. By combining a low-cost water-soluble chromophore with efficient analyte–probe interactions due to the use of aqueous solutions, the results here obtained provide a basis for designing sustainable sensing systems.

## 1. Introduction

The continuous growth of industrial and agricultural activities has led to an endless contamination and bioaccumulation of chemicals and toxic metals in the environment, which could cause serious health disorders on human health and could irreversibly damage the ecosystem’s equilibrium. The detection of toxic metal ions is particularly important for monitoring water pollution, food safety and disease diagnosis. For instance, exposure to lead ions can cause central nervous system dysfunction even at low concentrations [1]. On the contrary, copper ions, which are bioessential metal ions able to regulate various mechanisms in the human nervous system and the biochemical functioning of proteins [2], become toxic at high concentration, causing oxidative stress and neurodegenerative diseases [3]. Notably, metal ions such as Pb^2+^ and Cu^2+^ coexist in the environment, contaminating the atmospheric–soil–water distribution cycles and, because of their adverse effects on living organisms, simple sensing techniques are of particular interest in environmental monitoring. For this reason, the World Health Organization (WHO) has strictly defined the concentration limit of various metals ions that are allowed in water. Indeed, the maximum permissible limit of lead ions in water discharge is a concentration of 0.1 mg/L for waste water, whereas 0.01 mg/L (0.01 ppm) is permissible in drinking water. Regarding copper ions, the permitted level in drinking water is 1.3 mg/L (1.3 ppm).

Current methods for the detection of metal ions in water samples use sophisticated sensing techniques such as atomic absorption spectrometry (AAS) [4] and inductively coupled plasma mass spectrometry (ICP-MS) [5]. Both methods are extremely expensive, requiring a complex sample preparation and time-consuming analysis. Among different detection techniques, optical detection (fluorescent or colorimetric changes) is the most convenient tool because it is a simple method and generally shows high sensitivity, good selectivity and a fast response speed [6]. These sensors can contain a chromophore that specifically interacts with the target metal ions and translate this interaction to an optical response. To date, various optical probes for metal detection have been designed and investigated, including small molecules [7], calixarenes [8], polymers [9] and nanoparticles [10]. Although water-soluble receptors have already been reported on, the synthesis of such probes needs complicated synthetic pathways and/or the use of toxic/expensive starting materials [11,12]. To address these issues, stable chromophores with good water solubility, using simple chemical pathways and low-cost starting materials, are needed.

Perylene diimides (PDIs) represent an important class of materials, generally used as active components in organic solar cells [13,14], OLEDs [15] or interlayers [16] in optoelectronic devices, because of their outstanding features such as their high chemical and thermal stability, good absorption properties, excellent fluorescence and good electron-accepting capabilities [17]. These properties can be easily tuned with a molecular modification through a substitution at imidic or bay positions. While the chemical modifications leading to improved solubility in organic solvents are well-known and commonly used for the development of new materials for optoelectronic applications, there are only a limited number of reports dealing with structural modifications for imparting perylene derivatives with water solubility [18]. Since PDIs are highly hydrophobic, water solubility is usually achieved by introducing hydrophilic groups in the bay area or imidic positions. Moreover, the substitution of bay positions with charged substituents lowers or prevents the π–π interactions between the perylene planes, providing high fluorescence quantum yields in both the solid state and solution. With this approach, perylene diimides can be tailored for use as fluorescent labels in living cells or target-specific biolabeling [19]. On the other hand, the introduction of hydrophilic units at the imidic positions imparts water solubility while maintaining the tendency to aggregate via π–π interactions.

These remarkable features make PDI chromophores ideal molecular frameworks for the development of a broad range of sensors for the detection of heavy metal ions, inorganic anions or toxic organic compounds such as nitroaromatics, amines or nitriles [20]. In regards to lead ion detection, Wang et al. reported a PDI on–off fluorescent sensor with a detection limit of Pb^2+^ up to 10^−9^ M, although mixtures of DMF/water were used for the sensing experiments [21]. Yan et al. employed a turn-on probe based on the use of an aptamer and a water-soluble fluorescent perylene probe [22]. On the other hand, PDI-based chemosensors for Cu^2+^ often work in a dual mode, involving both colorimetric and fluorescent modulation, with the binding groups usually being amines attached at the imidic positions or bay area of PDIs [23,24]. In these cases, a good optical response was achieved upon the analyte–metal ion interaction in organic solvents or mixed organic/aqueous media. Qvortrup and coworkers [25] reported on an N-terminal-substituted PDI using a tetradentate ligand to detect Cu^2+^/Fe^3+^ in CHCl_3_/CH_3_OH solution, while Bojinov et al. synthesized a complex PDI chemosensor and studied its optical properties as a function of Cu^2+^ and Pb^2+^ ions [26].

Despite the good performances shown using these sensing systems, the above PDIs were obtained via complicated synthetic pathways or using co-analytes and co-solvents, therefore, enhancing the complexity of the sensor. Moreover, because of the strong tendency of aggregation through stacking interactions between the π-conjugated perylenic planes in aqueous media, most studied PDI-based chemosensors are developed in organic, organic–water or buffer media, in order to maintain the necessary molecular dispersion.

These studies show that there is a strong need to develop, with simple synthetic pathways, stable chromophores able to maintain a molecular dispersion state in water and generate a rapid and good optical response upon interacting with metal ions.

In this work, we report on a facile green one-pot synthesis of the potassium salt of 3,4,9,10-perylene tetracarboxylic acid (PTAS) by using commercially available perylene-3,4,9,10-tetracarboxylic acid dianhydride (PTCDA), water/ethanol as solvents and simple purification steps. The thus obtained PTAS is fully characterized and exploited as an optical probe for the detection of metal ions in aqueous media. The nature of the species responsible for the variation in the optical properties of the solutions is elucidated using NMR experiments. Our PTAS sensing system is cost-effective and selective to Cu^2+^ and Pb^2+^, showing an optical response in absorption spectra in water without the necessity to use co-analytes or organic solvents.

## 2. Results and Discussion

### 2.1. Photophysical Studies of PTAS

To assess the suitability of PTAS for sensing purposes, the spectroscopical behavior of the aqueous solution of PTAS at various concentrations was studied. All experimental measurements were conducted in distilled water (pH: 6.8), without the use of any organic solvents or buffers. In Figure 1, the absorption spectra of PTAS at different concentrations were compared. The absorption spectra displayed the typical features of perylene absorption, dominated by the vibronic progression of the first π–π* S_0_–S_1_ transition (peaks at 414, 439 and 466 nm) associated with the 0 → 2, 0 → 1 and 0 → 0 vibronic progression of the perylene ring modes at approximately 1380 cm^−1^ [27] and the higher energy S_0_–S_n_ transitions at 262 nm and 219 nm.

The well-resolved vibronic absorption bands together with the linear concentration dependence of the optical density and the constant ratio between the 0-0 and 0-1 absorbances (see inset of Figure 1) demonstrated that PTAS was molecularly dissolved in water in the 10–50 μM concentration range, showing a decadic extinction molar coefficient of 33485 M^−1^cm^−1^, in agreement with similar perylene derivatives [28].

The emission spectrum of PTAS in water displayed a good mirror symmetry with the absorption (Figure 2c), with the S_1_–S_0_ transition displaying peaks at 480, 511 and 548 nm (associated with the 0 → 0, 0 → 1 and 0 → 2 vibronic progressions of perylene ring modes of approximately 1300 cm^−1^) and a Stokes shift of 75 meV. The PL QY of 76% and lifetime of 4.45 ns did not show evident variations when modifying the concentration from 10 to 50 μM, again, confirming the condition of molecularly dispersed chromophore in this concentration range. We, therefore, assumed that the water PTAS solutions with concentrations up to 50 μM represented a suitable condition for using this dye in the detection of metal ions, since they did not display the evident formation of aggregated states. In particular, for cation sensing, we used a 30 μM PTAS concentration in water, which was the minimal concentration needed for naked-eye color detection (below this concentration, it is difficult to see the color of PTAS solutions with the naked eye).

### 2.2. Sensing Properties of PTAS for Metal Cations

Perylene chromophores are excellent probes for the detection of metal ions, but many detection methods reported in the literature have the disadvantage of using organic solutions or organic solvent mixtures, which have limited relevance (Appendix A). Most of these analytes are found in nature in aqueous media, and, therefore, their detection in water solutions is more reliable.

In this work, the sensing properties of PTAS for metal cations were studied by performing experiments in water, without the addition of other solvents or co-analytes. We also deliberately avoided the use of buffers to prevent the effect of the ionic strength and foreign species that could influence the interactions between the probe and the analyte.

#### 2.2.1. Optical Absorption

The sensitivity of PTAS to different metal cations (Mg^2+^, Ni^2+^, Fe^2+^, Hg^2+^, Cu^2+^, Zn^2+^, Co^2+^, Cd^2+^ and Pb^2+^) was investigated through absorption/emission spectroscopy. Indeed, the variations induced using the metal cations on the emission properties of PTAS water solutions were not indicative of metal detection, since we observed a reduction in emission intensity for all the metal cations. More interesting were the variations induced in the absorption spectra, as we noticed a reproducible increase in the absorption, at approximately 495 nm, specifically in the case of Cu^2+^and Pb^2+^, as depicted in Figure 3.

In Figure 4a,b, the absorption spectra of PTAS 30 μM solutions containing different concentrations of copper and lead analytes are reported. The optical variations were correlated with the concentration of metal cations and the time delay between their addition and the measure of the optical absorption. The variations in the absorbances, ΔA, induced with the metal cations at the absorption onset were evidenced in the insets, showing the growth of a band at approximately 505–510 nm. By analyzing these variations in more detail, we noted for all the metal cations a derivative modulation of the spectral shape in the 400–500 nm region, compatible with spectral broadening (second derivative, Appendix A) [29]. However, the band observed in the ΔA profile when adding Cu^2+^ and Pb^2+^ (Figure 4c,d) did not follow any derivative modulation of absorption and had to be associated with the formation of a new species in the solution, selectively induced with these two cations. This new species might be associated either with the formation of a metal complex or PTAS microaggregation. In the next section, NMR experiments were provided to clarify the origin.

The sensing capability of PTAS for Cu^2+^ and Pb^2+^ ions was assessed by performing UV-Vis spectroscopy titrations. We recorded the UV-Vis spectra of 30 μM solutions of PTAS containing different concentrations of copper or lead ions immediately after sample preparations to avoid a supramolecular assembly formation between PTAS and Cu^2+^ and Pb^2+^, which would give rise to aggregate precipitation. A detection limit of 2 μM was observed for both metals (see Figure 5a). As shown in Figure 5b, the use of water-soluble PTAS as the probe for the metal detection provided a simple visual optical method for the selective detection of two metal ions, being particularly relevant for monitoring the water environment.

#### 2.2.2. NMR Experiments

To analyze the binding mode of the chemosensor PTAS with Cu^2+^ and Pb^2+^, a series of ^1^H NMR experiments was recorded at different PTAS/metal ion ratios. A 30 µM stock solution of PTAS in deuterated water was analyzed and compared with those at different concentration of Cu^2+^. Trimethylsilylpropanoic acid (TSP) was used as the internal standard for the NMR experiments.

Before the addition of the Cu^2+^ ion solution, the ^1^H spectrum of PTAS evidenced only two groups of signals in the aromatic region, which were assigned to the H_α_ (at 8.43 ppm, *J* 8.04 Hz) and H_β_ (at 7.81 ppm, *J* 7.8 Hz) protons of the perylene unit (see Figure 6a). Successively, after the addition of various amounts of Cu^2+^, a new signal appeared at 8.46 ppm in the ^1^H spectrum of Figure 6a, and was assigned to a newly formed species with the carboxylic groups of the perylene units coordinated to Cu^2+^. Furthermore, the aromatic signals moved slightly to a low magnetic field, (0.0013 ppm shift), with a more pronounced effect for the H_β_ protons (0.0022 ppm shift), thus, suggesting a self-organization phenomenon for the noncoordinated species [30]. The phenomenon became more evident at higher Cu^2+^ concentrations as shown to have resulted from the broadening of the aromatic resonances in the ^1^H spectrum in Figure 6e [31]. The integral value of all species was evaluated through normalizing at 100 the integral value of H_α_ protons (but also with respect to the TSP signal as an internal reference for a doubled control) to monitor changes between the two species at increasing Cu^2+^ molar ratios. The results (Appendix A) indicated that at a low Cu^2+^ concentration (15 µM), the ratio between the coordinated and noncoordinated species was 1:2 and at Cu 60 µM was 1:1.4, to, finally, reach a ratio of 1:1 at Cu 120 µM.

The same NMR titration experiments were performed by adding an increasing amount of Pb^2+^ to the PTAS solution. After the addition of Pb^2+^ to the stock solution of PTAS, the ^1^H spectrum (Figure 7) changed and a new signal appeared at 8.46 ppm, together with the resonances of noncoordinated PTAS. The new signal was associated with the coordinated PTAS-Pb^+2^ species. Increasing the Pb^2+^ concentration, both the H_α_ and H_β_ protons (Figure 7), evidenced a downfield shift and a gradual decrease in intensity, suggesting that the noncoordinated perylene units were more perturbed with the presence of Pb^2+^. The different tendencies in the interactions between PTAS: Cu^+2^ and PTAS: Pb^2+^ were probably due to the different ionic radii that were 119 pm for Pb^2+^ and 73 pm for Cu^+2^. As a consequence, the more encumbered Pb^2+^ was, the better it coordinated the perylene units, perturbing the self-organizing tendency of free PTAS. The different coordination behavior of PTAS and Pb^2+^, with respect to PTAS and Cu^2+^, was evident when observing the resonances in the ^1^H spectra (Figure 7), which showed the signal of the noncoordinated perylene slowly decreasing at increasing Pb^2+^ concentrations. The integral value of all species in the ^1^H spectra was also evaluated for Pb^2+^ titration. Initially, at a very low Pb^2+^ concentration (PTAS:Pb^2+^ = 2:1), the coordinated and noncoordinated species were in a 1:2 ratio; increasing the concentration at 60 µM (PTAS:Pb^2+^ = 1:2), the ratio was 1:1.6; and at 120 µM (PTAS:Pb^2+^ = 1:4) was 3.9:1.

It is known from the literature that Pb^2+^ exhibits a higher coordination number with respect to Cu^2+^, giving rise to a hexagonal prism for Pb^2+^ and spherical aggregates for Cu^2+^ ions [32]. The higher coordination number for Pb^2+^ was the reason for the higher concentration decrease in noncoordinated PTAS and the consequent increase in coordinated PTAS/Pb^2+^, with respect to PTAS/Cu^2+^.

To determine the NMR detection limit, a series of titration experiments was conducted through ^1^H NMR concentration-dependent spectra. These spectra (reported in Figure 8 for Cu^2+^ and Figure 9 for Pb^2+^) displayed gradual changes in shape and chemical shifts in the aromatic resonances of PTAS at increasing metal concentrations. Even at very low concentrations, it was possible to detect the resonance associated with the interaction of PTAS with the metal ions and changes in noncoordinated PTAS signals, suggesting a self-organizing phenomenon even at very low concentrations.

## 3. Materials and Methods

### 3.1. Equipment and Experiments

The absorption spectra were recorded at room temperature using a Perkin Elmer UV/VIS/NIR Lambda 900 spectrometer. The spectra were recorded in quartz glass cuvettes and the extinction coefficients ε were calculated according to Lambert–Beer’s law.

PL spectra were recorded with a NanoLog composed of a iH320 spectrograph equipped with a Synapse QExtra charge-coupled device through exciting with a monochromated 450 W Xe lamp. PL QY was measured by using 4-(dicyanomethylene)-2-methyl-6-(4-dimethylaminostyryl)-4H-pyran in an ethanol solution as a reference, with an excitation wavelength at 443 nm. Time-resolved TCSPC measurements were obtained with a PPD-850 single-photon detector module and a DeltaTime series DD-405L DeltaDiode laser, and analyzed with the instrument Software DAS6.

The ^1^H-NMR spectra were recorded on a 600 MHz Bruker Avance spectrometer operating at 14.1 T, equipped with a 5 mm probe and gradient unit on z, and were then thermostated at 298 K (Bruker Biospin GmbH, Rheinstetten, Karlsruhe, Germany). Stock solutions in deuterated water of PTAS (300 µM), CuBr_2_ (100 µM) and Pb(NO_3_)_2_ were prepared. Samples were obtained through mixing the probe and analyte solutions with different ratios and bringing them to a final volume of 1 ml. Acquisition parameters for the ^1^H-NMR experiments: 90° pulse 8.25 μs PL1 -2.2 dB; relaxation delay 20.0 s; spectral width 8802.8 Hz; number of transient 64. For data processing, the exponential line broadening of 0.1 Hz was applied as the resolution enhancement function; zero-filling to 32 K prior FT, (TopSpin 4.0.6 software). As an internal standard, 20 μL of a 0.7 mM TMS/water solution was added to each NMR solution before data acquisition and spectra were referenced to the residual solvent signal of TMS at δ = 0 ppm. Spectra phasing and integration were performed manually.

### 3.2. Materials and Synthesis

The starting materials, perylene-3,4,9,10-tetracarboxylic acid dianhydride (CAS 128-69-8) and KOH (CAS 1310-58-3) were purchased from Sigma Merck and used without any additional purification.

PTAS was prepared using the alkaline hydrolysis of 3,4,9,10-perylene tetracarboxylic dianhydride. Briefly, 3,4,9,10-perylene tetracarboxylic dianhydride (2.55 mmol) was added to 200 mL ethanol and stirred at reflux temperature for 30 min. After this period of time, while refluxing, 200 mL 3% KOH aqueous solution was dropwise added. The color changed from a red suspension to a green solution. The reaction mixture was refluxed for an additional 12 h, then cooled at room temperature and added to the solution until the product was precipitated. The precipitate was filtered and dissolved in deionized (DI) water and added to ethanol. The pure product was filtered and washed several times with ethanol. Yield: 98%. The ^1^H-NMR (500 MHz, DMSO-d_6_): H_α_ (at 8.43 ppm, J 8.04 Hz) and H_β_ (at 7.81 ppm, J 7.8 Hz).

### 3.3. Metal Ion Detection

The metal ion solutions for the colorimetric sensor experiments were prepared by using distilled water with a pH of 6.8. Stock solutions of MgCl_2_, NiBr_2_, FeCl_3_, Hg(OAc)_2_, CuBr_2_, Zn(OAc)_2_, Co(NO_3_)_2_, Cd(NO_3_)_2_ and Pb(NO_3_)_2_ were prepared in a 100 μM concentration. A stock solution of PTAS with a concentration of 300 μM was also prepared.

Since the optimum concentration of PTAS to assure a good detection was established to be 30 μM, all the colorimetric studies were performed by combining the analyte–probe stock solutions to achieve a final concentration of 30 μM PTAS and different PTAS:analyte ratios.

## 4. Conclusions

A dual-responsive perylene-based chemosensor for detecting metal ions of Cu^2+^ and Pb^2+^ was synthesized at a low-cost with the water-soluble potassium salt of 3,4,9,10-perylene tetracarboxylic acid (PTAS). Selective and discriminative absorption variations were found for PTAS upon the addition of Cu^2+^ and Pb^2+^, with a detection limit as low as 20 ppm, reaching the sensitivity required for a copper analysis in drinking water and approaching that required for lead. The use of PTAS water solutions provided a simple visual optical method for the selective detection of two metal ions particularly relevant for monitoring water environments. NMR experiments evidenced, even at metal ion concentrations as low as 1 μM, the formation of new species associated with the interaction of PTAS with the metal ions. The different concentration behavior observed for the two metal ions correlated with their different coordination numbers.

Since the here reported data for aqueous environment were recorded at pH = 6.8, which was close to physiological conditions, these results constitute a promising platform for the exploitation of PTAS as active probes in chemo- (metal detection) and biosensing (enzyme activity monitoring) applications.

## Figures and Tables

**Figure 1 molecules-27-07079-f001:**
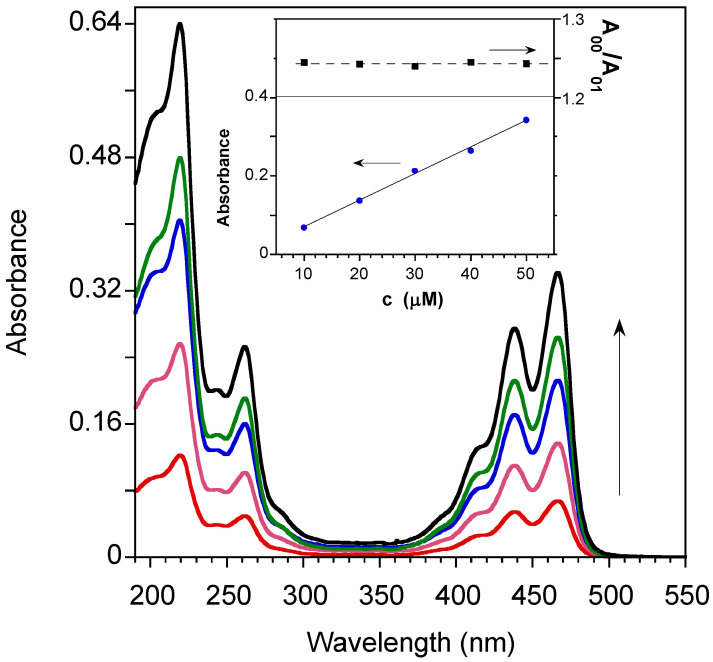
Concentration-dependent absorption spectra of PTAS in water (for 10, 20, 30, 40, 50 μM concentrations, increasing with the arrow). In the inset, the 0-0 S_0_–S_1_ absorbance (lower panel) and the absorbance ratio of the 0-0 and 0-1 peaks (upper panel) are reported as functions of the concentration for a 2 mm optical path cuvette.

**Figure 2 molecules-27-07079-f002:**
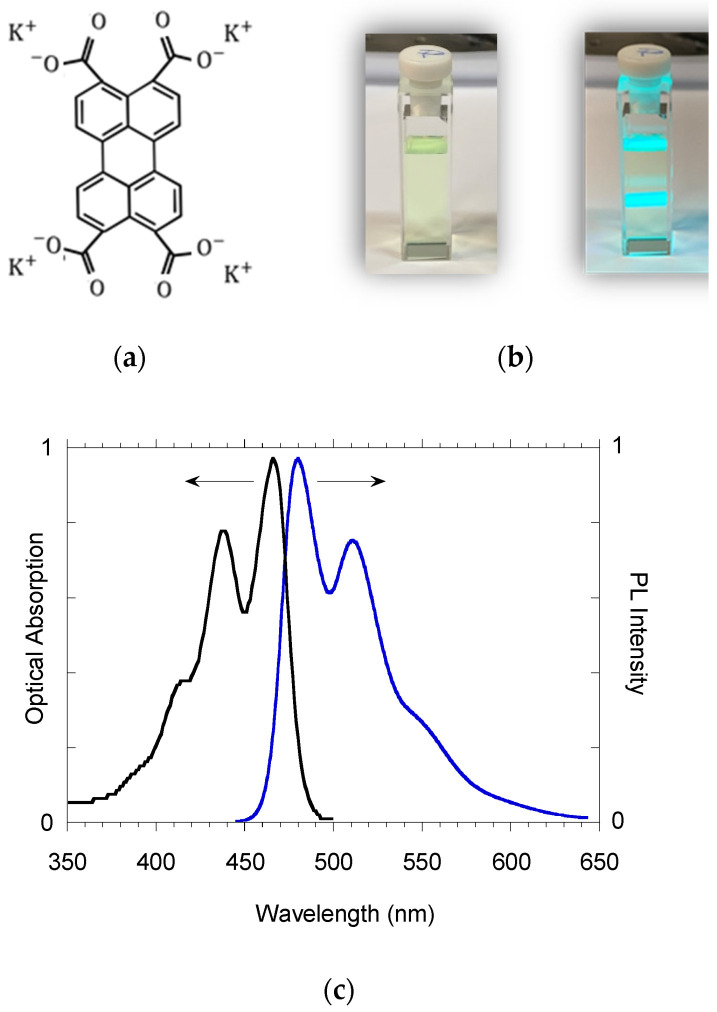
(**a**) PTAS chemical structure; (**b**) pictures of a PTAS water solution (c = 4 μM) in ambient light (left) and under excitation with a laser beam 408 nm (right); (**c**) absorption and emission spectra of PTAS in water (c = 4 μM).

**Figure 3 molecules-27-07079-f003:**
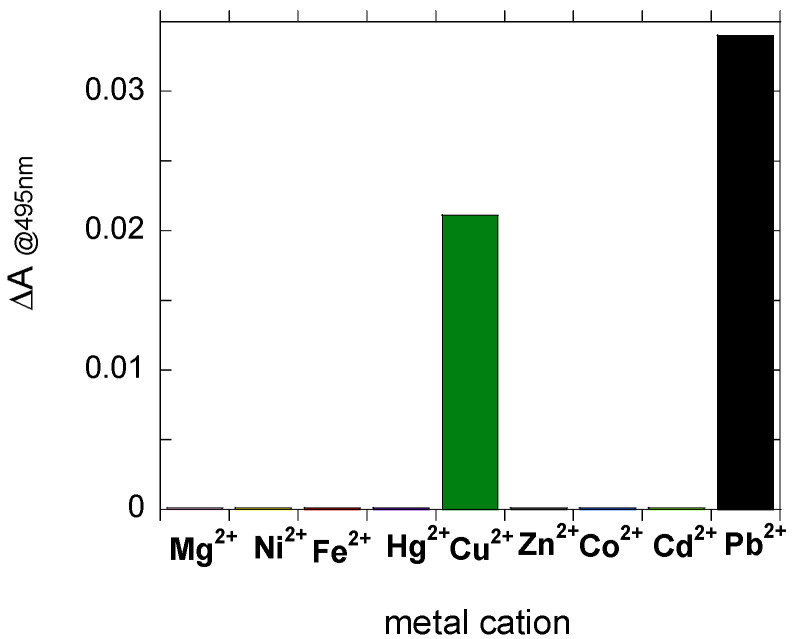
Variation of the absorbance measured at 495 nm with addition of 20 μM of different metal cations in PTAS aqueous solution of 30 μM.

**Figure 4 molecules-27-07079-f004:**
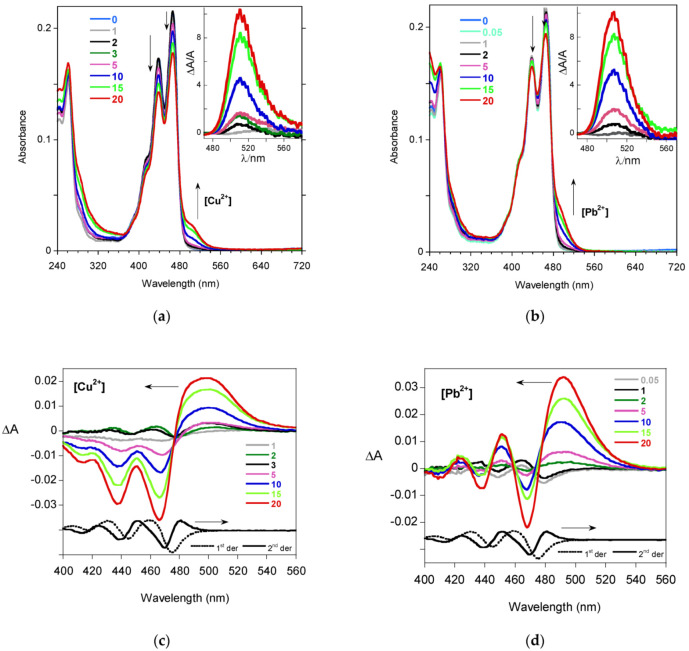
Absorption spectra of PTAS (30 μM) in water in the presence of (**a**) Cu^2+^ or (**b**) Pb^2+^, with different concentrations expressed in μM. In the insets, the relative absorbance variations ΔA/A are evidenced in the absorption onset region. The arrows indicate the optical variations when increasing the metal concentration. Absorbance variations ΔA, as a function of μM metal concentrations for (**c**) Cu^2+^ or (**d**) Pb^2+^, are compared to the first (dotted line) and second (solid line) derivatives of the PTAS absorption spectrum. Absorbance was measured for solutions in 2 mm optical path cells at room temperature.

**Figure 5 molecules-27-07079-f005:**
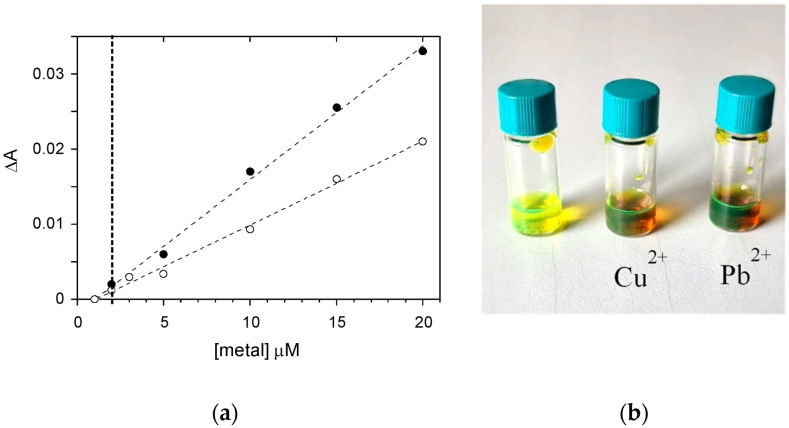
(**a**) Variation in the absorbance ΔA of a 30 μM PTAS water solution, measured at 490 nm, as a function of the metal ion concentration for Cu^2+^ (open circles) and Pb^2+^ (solid circles). The best linear fits are shown as dashed lines, the detection limit of 2 μM can be identified with the vertical bold dashed line. (**b**) Pictures of the PTAS water solution before (**left**) and after addition of 30 μM metal ions (**right**).

**Figure 6 molecules-27-07079-f006:**
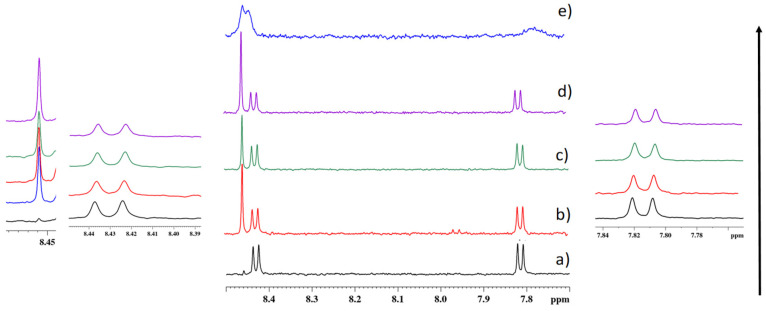
The ^1^H NMR spectra of PTAS at different metal ion ratios in D_2_O at 600 MHz. (**a**) A 30 µM PTAS solution; (**b**) 30 µM PTAS and 15 µM Cu^2+^ solution (PTAS:Cu^2+^ = 2:1); (**c**) 30 µM PTAS and 30 µM Cu^2+^ solution (PTAS:Cu^2+^ = 1:1); (**d**) 30 µM PTAS and 60 µM Cu^2+^ solution (PTAS:Cu^2+^ = 1:2); (**e**) 30 µM PTAS and 120 µM Cu^2+^ solution (PTAS:Cu^2+^ = 1:4).

**Figure 7 molecules-27-07079-f007:**
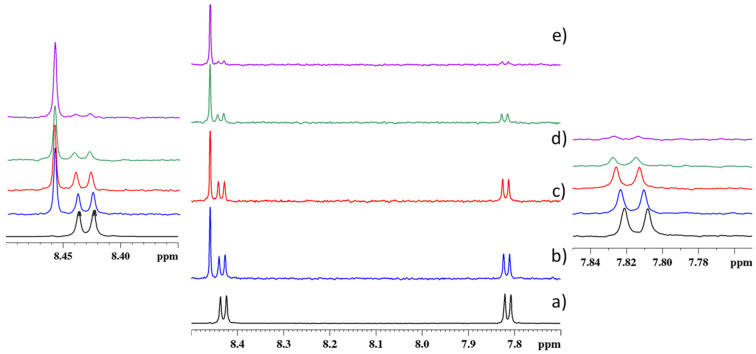
The ^1^H NMR spectra of PTAS at different metal ion ratios in D_2_O at 600 MHz. (**a**) A 30 μM PTAS solution; (**b**) 30 μM PTAS and 15 μM Pb^2+^ solution (PTAS:Pb^2+^ = 2:1); (**c**) 30 μM PTAS and 30 μM Pb^2+^ solution (PTAS:Pb^2+^ = 2:1); (**d**) 30 μM PTAS and 60 μM Pb^2+^ solution (PTAS:Pb^2+^ = 1:2); (**e**) 30 μM PTAS and 120 μM Pb^2+^ solution (PTAS:Pb^2+^ = 1:4).

**Figure 8 molecules-27-07079-f008:**
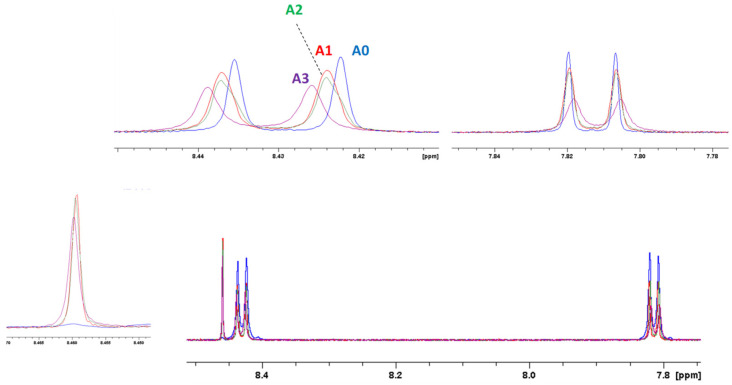
Concentration-dependent ^1^H NMR spectra of PTAS at different Cu^2+^ ion ratios in D_2_O at 298 K. A0: 30 μM PTAS solution; A1: 30 μM PTAS and 1 μM Cu^2+^ solution; A2: 30 μM PTAS and 2 μM Cu^2+^ solution; A3: 30 μM PTAS and 5 μM Cu^2+^ solution.

**Figure 9 molecules-27-07079-f009:**
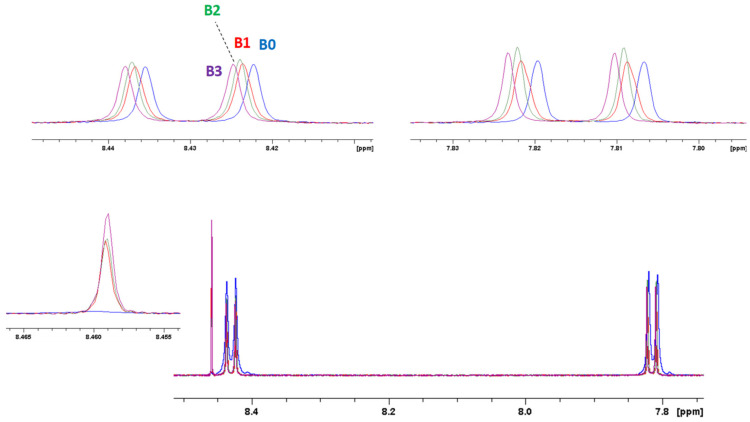
Concentration-dependent ^1^H NMR spectra of PTAS at different Pb^2+^ ion ratios in D_2_O, 298 K. B0: 30 μM PTAS solution; B1: 30μM PTAS and 1 μM Pb^2+^ solution; B2: 30 μM PTAS and 2 μM Pb^2+^ solution; B3: 30 μM PTAS and 5 μM Pb^2+^ solution.

## Data Availability

The data presented in this study are available in Appendix A.

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
