# Peer review of "Ultrasensitive Detection of Cu(II) and Pb(II) Using a Water-Soluble Perylene Probe"

_molecules, 2022, doi:10.3390/molecules27207079_

Round 1

Reviewer 1 Report

In this manuscript, the authors reported an ultrasensitive water-soluble perylene probe for detection of Cu2+ and Pb2+ in water. The probe PTAS exhibits low detection limits for both Cu2+ and Pb2+, and provides a simple visual optical method for the selective detection of two metal ions particularly relevant for monitoring water environment. This work is meaningful, and can be published in my opinion. Following are some minor comments.

1. Please pay attention to the writing of ions in the paper.

2. What is illustrated in Figure 2(b), please add relevant descriptions.

3. As shown in Figure 5(b), after adding Cu2+ and Pb2+, the color of solution significantly changed. Please supplement the fluorescence spectra of PTAS in water in the presence of Cu2+ or Pb2+, and the fluorescence responses of PTAS to various metal cations.

4. In NMR experiments, the binding mode of PTAS with Cu2+ and Pb2+ was discussed, please provide a diagram of the binding mode.

5. It is better to use a table to compare this work with other reported probes for the Cu2+ or Pb2+ detection.

Author Response

We thank the reviewer for the positive evaluation of the manuscript and for the helpful comments that help us to improve it,

here is the point-by-point response to his/her comments:

  1. Please pay attention to the writing of ions in the paper. 

We have now corrected accordingly

  1. What is illustrated in Figure 2(b), please add relevant descriptions.

The caption of Figure 2(b) has been changed to better describe the  reported pictures

  1. As shown in Figure 5(b), after adding Cu2+and Pb2+, the color of solution significantly changed. Please supplement the fluorescence spectra of PTAS in water in the presence of Cu2+ or Pb2+, and the fluorescence responses of PTAS to various metal cations.

As explained in the manuscript (lines 157-159) the variations induced by the metal cations on the emission properties of PTAS water solutions are not indicative for metal detection since only a reduction in emission intensity for all the metal cations is observed, mainly attributed to PTAS precipitation.

  1. In NMR experiments, the binding mode of PTAS with Cu2+and Pb2+ was discussed, please provide a diagram of the binding mode.

The binding mode of PTAS with Cu2+ and Pb2+ can be evaluated from changes in the chemical shifts of the 1H resonances of coordinated and non-coordinated species, and from the titration experiments. Furthermore, a Figure was added in the SI showing the integrals value referring to the titration experiments and evaluated at several PTAS:Cu2+ ratios.

Reviewer 2 Report

This paper studied the detection of Cu(II) and Pb(II) using potassium salt of 14 3,4,9,10-perylene tetracarboxylic acid (PTAS) as the probe. The detection limits for Cu(II) and Pb(II) are about 1 uM using NMR and 2 uM using UV. However the specificity to detect Cu(II) and Pb(II) need to be improved. The absorbance difference can be barely distinguised from other metal ions.

Line 160, "about 500 nm" should be "at 495 nm".

Line 163, Variation of the absorbance measured at 495 nm is within 0.03 difference. This could be due to experimental error also. Thus I don't think this is a practical and acurate way to detect Cu(II) and Pb(II). 

Line 200, "1H NMR" should be "1H NMR"

Line 284, "CuBr2 (100 µM) and Pb(NO3)2" should be "CuBr2 (100 µM) and Pb(NO3)2

Line 306, "1H-NMR" should be "1H-NMR"

Author Response

*point 5 of Reviewer 1 (not included in the previous answer)

  1. It is better to use a table to compare this work with other reported probes for the Cu2+or Pb2+ detection.

As suggested by the referee 1 we have added a Table in the Supporting Information where perylene probes for lead and cupper ions detection are compared.

We thank the reviewer 2 for the positive evaluation of the manuscript and for the helpful comments that help us to improve it,

here the point-by-point response to his/her comments:

This paper studied the detection of Cu(II) and Pb(II) using potassium salt of 14 3,4,9,10-perylene tetracarboxylic acid (PTAS) as the probe. The detection limits for Cu(II) and Pb(II) are about 1 uM using NMR and 2 uM using UV. However the specificity to detect Cu(II) and Pb(II) need to be improved. The absorbance difference can be barely distinguised from other metal ions.

Line 160, "about 500 nm" should be "at 495 nm".

The wavelength has been corrected

Line 163, Variation of the absorbance measured at 495 nm is within 0.03 difference. This could be due to experimental error also. Thus I don't think this is a practical and acurate way to detect Cu(II) and Pb(II). 

The variation of the absorbance measured at 495 nm is reproducible and it occurs only when Cu(II) and Pb(II) is added to the solution, even if te variation is small, it can not be an experimental error. As shown in Figure 3, with a simple UV-Vis spectrometer this 0.03 variation in absorbance is easily detected, we therefore believe that this method can be successfully applied to detect metal ions.

Line 200, "1H NMR" should be "1H NMR"

now corrected

Line 284, "CuBr2 (100 µM) and Pb(NO3)2" should be "CuBr2 (100 µM) and Pb(NO3)2

now corrected

Line 306, "1H-NMR" should be "1H-NMR"

now corrected

Reviewer 3 Report

Kozma et al. molecules-1943236

In this manuscript, the authors report sensitive sensing of Cu(II) and Pb(II) in water using a perylene tetracarboxylates (PTAS). The complexations were characterized by absorption spectroscopy as well as proton NMR spectroscopy both quantitatively and qualitatively. The PTAS shows ultrasensitivity toward binding of Cu(II) and Pb(II) in water that underscores a cheap and easy way for detecting trace amounts of these toxic heavy metal ions.

A few comments:

(1)    In the introduction, the author discussed that “Still, most of receptors used for sensing lack of water solubility”. The sensing of heavy metal ions in water by either water-soluble dyes, nanoparticles or polymer-based materials are well-developed over the years. The authors need to cite a few literatures in these fields. I listed a few here for sensing Cu(II) and Pb(II) in aquous solution just FYI.

a)       Talanta 2012, 94, 172– 177,

b)      J. Sensors., 2016, Article 1905454.

c)       Chem.Eur.J.2016,22,3037–3043.

d)      ACS Macro Lett. 2019, 8, 1, 79–83.

e)      Spectrochim. Acta, Part A, 2019, 223, 117348.

f)        Spectrochim. Acta, Part A, 2020, 239, 118485.

(2)    The authors need to make sure to use “1H NMR” with superscription on 1.

(3)    Where are the data the authors mentioned on line 219 for the integrals for the NMR titration experiments? The authors are encouraged to put them in the SI.

Summary,

This manuscript is recommended for publication in Molecules after minor revisions.

Author Response

We thank the reviewer for the positive evaluation of the manuscript and for the helpful comments that help us to improve it,

here the point-by-point response to his/her comments:

(1)    In the introduction, the author discussed that “Still, most of receptors used for sensing lack of water solubility”. The sensing of heavy metal ions in water by either water-soluble dyes, nanoparticles or polymer-based materials are well-developed over the years. The authors need to cite a few literatures in these fields. I listed a few here for sensing Cu(II) and Pb(II) in aquous solution just FYI.

  1. a)Talanta 2012, 94, 172– 177,
  2. b) Sensors., 2016, Article 1905454.
  3. c)Eur.J.2016,22,3037–3043.
  4. d)ACS Macro Lett. 2019, 8, 1, 79–83.
  5. e) Acta, Part A, 2019, 223, 117348.
  6. f) Acta, Part A, 2020, 239, 118485.

We thank for te helpful suggestion, we have added some of the References above, as suggested from the reviewer

(2)    The authors need to make sure to use “1H NMR” with superscription on 1.

Now corrected

(3)    Where are the data the authors mentioned on line 219 for the integrals for the NMR titration experiments? The authors are encouraged to put them in the SI.

A Figure was added in the SI (namely, Figure S2). In the Figure, a proton experiment with the integrals value is illustrated as an example. Moreover, in the figure a table is added showing the integrals value referring to the titration experiments and evaluated at several PTAS:Cu+2 ratios.